# Main Factors Influencing Whole Grain Consumption in Children and Adults—A Narrative Review

**DOI:** 10.3390/nu12082217

**Published:** 2020-07-25

**Authors:** Alexandra Meynier, Aurélie Chanson-Rollé, Elisabeth Riou

**Affiliations:** 1Mondelez France R&D SAS, 91400 Saclay, France; elisabeth.riou@mdlz.com; 2VAB-Nutrition, 63100 Clermont-Ferrand, France; aurelie.chanson-rolle@vab-nutrition.com

**Keywords:** whole grains, intakes, barriers, facilitators, children, adults

## Abstract

Despite their recognized health benefits, intakes of whole grains (WG) are below recommended levels in almost all countries worldwide. This observation highlights the need to increase WG consumption by understanding factors influencing this consumption and how they could be favorably impacted. This review focused on facilitators of and barriers to WG consumption and how to improve the effectiveness of programs aiming at increasing WG consumption. The main methods to facilitate WG intakes in both adults and children seem to be to (i) increase the availability and the variety of foods containing WG, (ii) improve their sensory appeal, (iii) reduce their purchase cost, (iv) use a familiarization period to introduce them to consumers (with a gradual increase in consumed amounts and repeated exposure), and (v) improve communication and labeling to enhance consumers’ ability to identify products with WG. These strategies may be used to improve the effectiveness of programs aiming at promoting WG consumption, with a further emphasis on the need to apply them over a long period of time, and potentially to include tasting sessions of new foods containing WG. Finally, these strategies should involve broad partnerships between multiple stakeholders at the regulatory, institutional and industrial levels.

## 1. Introduction

Whole grain (WG) consumption has been shown to reduce the risk of several non-communicable diseases, such as cardiovascular diseases, type 2 diabetes and some types of cancer [1,2]. However, few countries have integrated quantitative recommendations for WG in their dietary guidelines. When such recommendations exist, they go from 48 g/d (three servings) in the United States to up to 90 g/d for men in Sweden and Norway. Other countries have qualitative (i.e., descriptive) and non-specific recommendations based on “increasing” consumption of WG or “choosing” preferentially WG options [3,4] (see Table 1).

However, actual WG intakes are below these recommendations in almost all countries worldwide. In children and adolescents, data from nationally representative surveys show that average intakes (expressed in amounts of WG ingredients) range from approximately 2 g/day in Malaysia and Italy to 23 g/day in Ireland, and up to 58 g/day in Denmark (Figure 1). In adults, they range between about 4 g/day in Italy and 28 g/day in Ireland, and reach 58 g/day in Denmark (Figure 2). In the USA, the average intake of WG in the overall adult population is around 15 g/day. Therefore, there seems to be a need to increase WG consumption in both children and adults, and to understand what factors may influence WG intakes in these populations. Several studies have addressed the consumer’s perception of products made with WG in order to try to identify the main influencing factors of WG consumption in various age groups (e.g., [15,16,17]). Nevertheless, few reviews have tried to collate and synthesize data obtained from these studies in both children and adults, while such analyses would be needed to identify the most relevant factors among those already identified in the literature. Furthermore, several programs or interventions have tried to increase WG intakes through leveraging some of the factors identified as influencing WG consumption, with various levels of efficacy. It would therefore be useful to summarize the information available from these studies in order to identify the main reasons for failure or success of the implemented programs. Evaluating the impact of programs will also make it possible to confirm, through an objective criterion (measurement of actual WG consumption), what would be the most effective influencing factors among those identified in consumer’s perception surveys. Overall, undertaking this research will help to ascertain what could be the most effective strategies that should be implemented in order to beneficially and effectively influence WG consumption.

In this context, we performed a review of the literature with the objectives of: (i) synthesizing, ranking and better understanding the factors influencing WG consumption (barriers and facilitators) in both children and adults, and (ii) identifying the reasons for success or failure of programs that have tried to impact WG consumption in both children and adults in order to confirm more objectively the efficacy of these factors. Studies have been identified through a search on the MEDLINE database (that has been accessed through PubMed^®^ Rockville, Maryland, USA), by combining WG-related keywords [e.g., “whole(-)cereal(s)” OR “whole(-)grains” OR individual cereal names such as “wheat” OR “rye”] to the following set of keywords: “consumer(s) OR perception(s) OR barrier(s) OR facilitator(s) OR program(me)(s) OR intervention(s)”. This primary search was completed by a snowball strategy that consisted in seeking for any relevant studies within the list of references of analyzed articles. Two types of studies were selected for our review, namely studies addressing the consumer’s perception of WG, and studies describing the efficacy of programs aimed at promoting WG consumption. Furthermore, data were considered separately for the following age groups: children (3–12 years), adolescents (12–18 years), young adults (18–30/40 years), middle-aged adults (30/40–60/65 years) and older adults (>60 years).

## 2. Main Barriers to and Facilitators of Whole Grain Consumption

Information regarding the factors that may influence WG consumption (barriers and facilitators) was collected from studies addressing the consumer’s perception of products made with WG. The literature search allowed us to identify 13 studies in children [16,31,32,33,34,35,36,37,38,39,40,41,42], five in adolescents [33,43,44,45,46], and 30 studies in adults; more precisely for adults: 10 in young adults [44,47,48,49,50,51,52,53,54,55], 10 in middle-aged adults [16,31,37,55,56,57,58,59,60,61], four in older adults [17,62,63,64], two in young and middle-aged adults grouped together [65,66], and five that did not separate age groups and considered all adults together [15,67,68,69,70]. The data that were used for the review concerned the facilitators of and barriers to WG consumption that were directly mentioned by the evaluated subjects themselves. For children, this information was sometimes collected from their care-givers (parents in most cases) instead of from the children themselves. These data have been extracted together with information regarding the study experimental design (e.g., interventional study or focus group evaluation with or without inclusion of a tasting session), the population’s characteristics (age, health status, gender and sample size), and the country where and the year(s) when the study was performed. The factors (barriers and facilitators) identified in each study were listed and grouped in consistent and homogenized categories to allow for comparisons between studies, as shown in Figure 3. For instance, the factor “improvement of sensory appeal” corresponds to the improvement of the sensory characteristics of the WG-containing products that are made available to consumers, while the factor “preference/liking of taste/texture” corresponds to the consumers’ established positive perception of the taste/texture of WG-containing foods that are already on the market. Conversely, the factor “dislike of taste/texture” corresponds to the consumers’ established negative perception of the taste/texture of available WG-containing products. The factor “lack of appeal (appearance/pack/marketing)” encompasses the low attractiveness of foods containing WG for consumers in a global manner and before consumption, which may include aspects related to color, packaging and marketing. Furthermore, the term “availability” refers to the frequency of the presence of WG-containing products in stores and other relevant locations (e.g., restaurants, school canteens…), while the term “variety” refers to the different types of WG-containing products that can be found (e.g., pizza, bread, muffin…). In regard to the latter, increasing variety would therefore imply increasing the number of types of existing products, in order to more widely address the different dietary habits of consumers. Finally, the factor “identify WG-containing products” corresponds to the ability of consumers to recognize foods containing WG, especially in comparison to foods containing refined grains. It is identified as a barrier when this ability is considered to be poor (“difficult to identify WG-containing products”), while improving this ability is considered as a facilitator. Once grouped in homogenized categories, the identified influencing factors (barriers and facilitators) were sorted depending on their relative importance for each age group separately. This ranking was based on the number of individual studies that have concluded that the corresponding factor was a barrier or a facilitator (see Figure 3).

A summary of the data collected from the identified and analyzed studies (as described above) is presented in Figure 3 for each of the five age groups of interest. These data show that the main methods to facilitate WG consumption in all age groups would be to increase the availability and the variety of foods containing WG, to improve their sensory appeal and their organoleptic properties (taste, texture and appearance), to reduce their purchase cost and to improve the labeling, communication and knowledge regarding WG in order to enhance the subjects’ ability to identify WG-containing foods. As shown in Figure 3, some of these elements, such as acting on the sensory characteristics and availability of WG-containing products, appear to be equally important in all age groups, while others display varying degrees of importance depending on age. For instance, cost is considered as a major issue in adults, especially in older ones, but seems to be less important for adolescents, and is not even mentioned at all for children. Similarly, the need to improve the ability to identify WG appears to be a more significant issue in adults, especially for the older ones, than in children and adolescents.

In regard to sensory perception, several studies show that adult subjects have an a priori negative image of WG-containing products, and that this can be improved with tasting and familiarization with these products [49,54,65]. As an example, in Neo et al. 2017, adult consumers indicate that foods containing WG take longer to chew and have a grainy texture and a floury taste [54]. One other aspect that may discourage the consumption of WG-containing products is the brown color, which is felt to be a marker of inferior quality [54]. Muhihi et al. 2013 [59] also report that the perception of foods containing WG tends to be better in women compared with men. In regard to the ability to identify foods containing WG, efforts should be made to improve the clarity of WG labeling, and to avoid the use of overwhelming information and of vague and non-specific cereal-related terms (e.g., “multigrain,” “wheat,” or “stone-ground”), in particular for WG-containing foods that are aimed at older adults [57,64].

An additional facilitator that seems to be relevant for all age groups, except for adolescents and older adults, would be to apply a familiarization period to introduce products made with WG to consumers. This could be achieved through a gradual increase in the amount of WG ingredients contained in cereal products by progressively replacing refined grain ingredients with WG ingredients (what we have called the “small change” approach), combining WG and refined grain foods in daily menus, repeated exposure to foods containing WG, or the distribution of WG intakes throughout the day. We will present in the next section some studies that evaluated the efficacy of such familiarization periods within programs aiming at improving WG intakes (e.g., [71] or [72] in children, and [73] in adults).

Besides, some of the influencing factors that have been identified would be relevant only for particular age groups. This would be, for instance, the case for the provision of education on how to prepare and cook foods containing WG, which would further encourage WG consumption in adults only, although this would concern adults of all ages. In younger subjects, a specific strategy could be to favor the incorporation of WG ingredients into foods that are already habitually consumed and well-liked by these populations. Finally, consideration of the convenience of the food (i.e., ease and speed of preparation and consumption) should also be taken into account for teens and younger adults specifically, while the existence of chewing difficulties should be considered for older adults (see Figure 3).

## 3. Main Reasons for Success and Failure in Programs to Promote WG Consumption

Information regarding the reasons for success or failure of programs that aimed to promote WG consumption was obtained from studies describing the efficacy of such programs in children and adults. This analysis was performed in order to collect data regarding the common characteristics of effective (reasons for success) and ineffective (reasons for failures) programs by using an objective criterion of evaluation, which was the measured subjects’ WG intakes. More precisely, we included in our analysis all relevant information in relation to the description of the programs identified, i.e., (i) age group(s), (ii) settings (environment where the program was implemented, e.g., at home, at school…), (iii) whether the program was focused on WG specifically or also aimed at promoting other types of foods or healthy behaviors in general, (iv) components that were used to promote WG consumption (see Table 2 and Table 3 for examples), (v) population targeted by the program (e.g., the children themselves or their care-givers), and vi) type of program (defined as governmental, industrial or academic/research) in relation to the entity (ies) instigating the program or that provided financial support for the research. Furthermore, we also considered the conclusions regarding the impact of the tested program or intervention on the subjects’ WG intakes (i.e., consumed amount or frequency of consumption), and we used these conclusions to determine whether a program/intervention was effective (i.e., induced a significant increase in WG intakes) or not (i.e., induced no significant changes in WG intakes). We then identified the common characteristics between effective studies as compared to non-effective studies in order to uncover the possible main reasons for the success or failure of programs. When available, information regarding the reason(s) for success or failure of the tested programs as identified by the investigators themselves was also considered in our own analysis. Finally, the identified reasons for program success/failure were sorted depending on their relative importance for each age group separately, on the basis of the number of studies that have concluded that the corresponding factor was a reason for success or failure.

The literature search allowed us to identify 21 programs in children and adolescents; more precisely, nine programs in children (described in 12 studies [71,74,75,76,77,78,79,80,81,82,83,84]), seven in adolescents [46,85,86,87,88,89,90], and five that did not separate age groups and considered children and adolescents together (described in nine studies [18,72,91,92,93,94,95,96,97]). For adults, 10 programs were identified: four in young adults (described in six studies [49,76,98,99,100,101]), two in middle-aged adults [57,74], two in older adults [62,63], one that considered young and middle-aged adults together [73], and one that considered all ages together (described in three studies [18,97,102]).

A description of all identified programs is presented in Table 2 and Table 3, for children and adults, respectively. These tables also show conclusions regarding the impact of the program on the subjects’ WG intakes as measured throughout each study. As illustrated in these two tables, most of the identified programs (22 programs out of 31 when considering all age groups) have been performed in the United States, and only 13 programs out of these 31 were specifically focused on the promotion of WG consumption. In children and adolescents, the majority of the programs had a school-based setting, and in adults, home or community-based settings were more common, followed by university-based settings for young adults specifically. Finally, the programs in children and adolescents were of various types, with the highest prevalence emanating from academic researchers, and the remainder relatively balanced between private programs (supported by non-governmental organizations or industries) and public or governmental programs. Some programs involved both public and private stakeholders, such as the Fuldkorn program in Denmark [18,97,102]. In adults, almost all identified programs were performed by academic researchers (see Table 2 and Table 3 for details).

A summary of the data collected is described in Figure 4. These data show that the main factors that increase the chance of success for a program to favorably impact WG consumption in all age groups seem to be led by the introduction of WG within a large variety of foods that are habitually consumed by the subjects, over a long period of time with repeated exposure to foods containing WG. Tasting sessions of the WG-containing products should also be included. Programs should also include the provision of nutritional education in relation to WG, using materials that are practical, simple, interactive, and focused on key messages related to the identification of products containing WG and to their importance for health and disease prevention. Additional levers that would improve the chance of success in programs targeting children and adolescents specifically would be the implementation of a “small change” approach for the introduction of WG into cereal products (as defined before), the involvement of parents in the process, and the use of social marketing approaches (i.e., approaches using commercial marketing tools and principles to try changing health and social behaviors). Nudging (i.e., paying attention to the way foods containing WG are presented) may also be of interest for younger children. For programs aimed at adults, specific strategies to enhance the programs’ chance of success in increasing WG consumption would be to deliver education on how to prepare and cook foods containing WG and improve label reading skills, as well as a strong involvement from government and non-governmental organizations to provide support for such programs. Furthermore, addressing the subjects’ motivation to change their diet and the level of availability of foods containing WG in stores may be other ways to contribute to the programs’ efficacy, although these strategies were less often highlighted in the studies we identified.

A last factor that would be important for all age groups seems to be the implementation of programs through a broad partnership that would involve both public stakeholders (regulatory and institutional agencies) and private stakeholders (industries). Indeed, implicating different types of stakeholders would help to better address the main facilitators of WG consumption identified in Section 2, which would improve the chance of success of a WG program. More precisely, industries would have a key role to play to increase the availability and variety of WG-containing products, as well as to improve their sensory appeal and to formulate them according to the small change approach (with a gradual increase in incorporated WG amounts). Public stakeholders would have a role in encouraging industries to act as described above, and to engage reflections regarding products’ cost. They could also work in concert with industries to try to implement homogenized and clear WG labeling systems in order to improve the ability of consumers to identify WG-containing products. Finally, public health authorities could collaborate with non-governmental organizations (NGO) in order to encourage consumers to obtain and eat WG-containing products. This may include joint efforts to include clear quantitative recommendations regarding the amount of WG to be consumed within national dietary guidelines, or to implement public health campaigns on the promotion of WG. A typical example of success for such broad partnerships is the “Fuldkorn program” in Denmark. It was built as a partnership across different sectors and disciplines such as health and patient organizations, industries, government, retail and trade. The main objectives were to improve the accessibility and identification of WG-containing products, as well as the awareness of beneficial effects of WG. This program made it possible to increase WG consumption from 33 g/d in 2000–2004 to 58 g/d in 2011–2013 in adults, and from 28 to 58 g/d in children over the same period [18,97,102] (see also Table 2 and Table 3).

Finally, in regard to the programs’ setting, it seems that in children and adolescents industry-based and school-based programs may have a higher chance of success than home-based programs, provided appropriate training and education of school food service staff is given, while in adults industry-based and home-based programs seem to be more effective when compared with university or institution-based programs. However, these latter conclusions should be treated with caution, given the small number of studies identified with an industry-based setting or an institution-based setting.

## 4. Discussion

Despite the recognized nutritional and health benefits of WG, intakes of WG are below recommended levels in almost all countries worldwide. This observation highlights the need to increase WG consumption in almost all populations, and to improve the understanding of the factors that are influencing WG consumption, and how they could be favorably impacted.

To our knowledge, the current review has been the first to collate information about the facilitators of and barriers to WG consumption coming from studies on consumer perception. The data collected show that the most effective ways to facilitate WG intakes in both adults and children would be to increase the availability and the variety of foods containing WG, improve their sensory appeal, reduce their purchase cost, use a familiarization period to introduce WG to consumers (with incorporation of gradually increasing amounts into cereal products and repeated exposure), and improve the labeling, communication and knowledge in relation to WG in order to expand the consumers’ ability to identify WG. Although not surprising, these elements correspond to the facilitators of WG consumption that have been most often highlighted by consumers in the identified studies. The above strategies may improve the chance of success of programs aiming at promoting WG consumption in children and adults, but need to be applied consistently over a long period of time. Such programs should also include tasting sessions of the foods containing WG, which may have been newly developed, to increase their acceptance by consumers. Furthermore, for children and adolescents, programs to improve WG consumption that are carried out in a school-based environment seem to be more effective than those achieved in a home-based environment, while in adults, home-based programs seem to be the most successful. Moreover, in all age groups, interventions that would involve industries to impact on the overall offer of WG-containing foods would be helpful to favorably influence consumers’ intakes of WG, although the literature identified a lower number of such programs, especially in adults. Finally, it seems that the implementation of broad partnerships involving both public stakeholders (regulatory and institutional agencies) and private stakeholders (industries) would be key to increasing the chance of success of programs intended to promote WG consumption in all groups of consumers.

The topics covered in the current review have been poorly addressed in the past. A review published in 2018 by Suthers and coll [103], which focused on public health interventions aimed at increasing WG intakes, drew approximately the same conclusions as us regarding the characteristics of studies that would be effective to improve WG consumption. Furthermore, there has been limited information from the literature regarding how the identified strategies to facilitate WG intakes in both adults and children could contribute to the extreme variations in WG consumption that have been observed depending on countries (as illustrated in Figure 1 and Figure 2). This is because programs that would efficiently impact on these different aspects, or even on some of them only, have been applied in a small number of countries. The best available example in the literature comes from Denmark which is, to our knowledge, the sole country to have implemented a nation-wide program addressing all the main facilitators of WG consumption identified in this review, through the so called “Fuldkorn program”. As described in Table 2 and Table 3, this program, which has been involving both public and industrial stakeholders, has led to a substantial increase in WG intakes of the Danish population. These results also further highlight the importance of the implementation of broad partnerships to successfully impact on several if not all facilitating factors of WG consumption [18,97,102].

Interestingly, policy makers with an interest in the promotion of WG should be able to use the data described in this review (as summarized in Figure 3 and Figure 4 and in Table 2 and Table 3) to find relevant information for their own country, and to appreciate the relative importance of the different factors influencing WG consumption when considering the consumers’ point of view (Figure 3), and when considering a more objective evaluation criterion focused on WG intake quantification (Figure 4). They would also be able to elaborate on our own conclusions regarding the main influencing factors identified to implement appropriate strategies with all relevant stakeholders (public and private) for their own country. More specifically, we believe that the possible practical implications of this review towards policy making may be diverse. First, our conclusions may encourage policy makers to implement a clear and homogenized WG labeling system, to include quantitative recommendations of WG consumption in national dietary guidelines, and to organize public health campaigns to communicate on WG towards consumers (focusing on the provision of clear recommendations of consumption, tips for their identification and information regarding their benefits). They could also motivate policy makers to find ways to encourage industries to incorporate more WG in their products. As an example, this may include granting the possibility for industrials to communicate on the achieved efforts regarding the reformulation of products with WG through the use of a WG logo, as it has been implemented in the frame of the Fuldkorn program in Denmark [18,97,102]. Policy makers may also be interested in exploring strategies to try to reduce the cost of WG, especially for low income families (e.g., through public aid programs).

The strengths of this review rely on the fact that we have identified the main factors influencing WG consumption in children and adults by collating data both from studies addressing the consumer’s perception of WG, and from studies describing the efficacy of programs aimed at promoting WG consumption. Data from the second type of studies made it possible to confirm conclusions derived from the first type of studies through the consideration of a more objective evaluation criterion (i.e., actual WG consumption as measured throughout the identified studies). Nevertheless, our review also has some limitations, which may affect the interpretation of the results and should therefore be acknowledged. Firstly, the literature search was performed through the Medline database only, which may have led to skipping some relevant studies. However, we tried to overcome this limitation through the use of a snow ball search strategy that allowed the identification of a substantial number of additional studies. Secondly, consumer perception data are by definition subjective, and may therefore not necessarily reflect reality. As explained above, we aimed to tackle this issue by including data from studies on programs that included an objective measure of WG consumption. Thirdly, it should be noticed that there was substantial heterogeneity in the design of the studies identified for our review, which could include focus group discussions with or without tasting sessions as well as observational or interventional studies with questionnaires regarding perception of WG. This may have biased some of the conclusions derived from individual studies. Finally, few investigations have been retrieved for particular WG-containing products. This prevented us from drawing specific conclusions depending on the types of foods, which would have been a valuable source of information for policy makers and other public and private stakeholders.

## 5. Conclusions

The information described in this review will be valuable to develop new strategies, at both the public health and industry levels, to increase WG consumption in children and adults. These strategies should focus on increasing the availability and variety of foods containing WG, and should involve broad partnerships between multiple stakeholders at the regulatory, institutional and industrial levels. Finally, clear and harmonized recommendations of WG consumption, as well as clear definitions of what could be considered WG ingredients and foods containing WG are needed. These latter aspects would help consumers to identify food products that may contribute meaningfully to improving their daily consumption of WG, in order to reach the daily amounts needed to favorably impact on their health.

## Figures and Tables

**Figure 1 nutrients-12-02217-f001:**
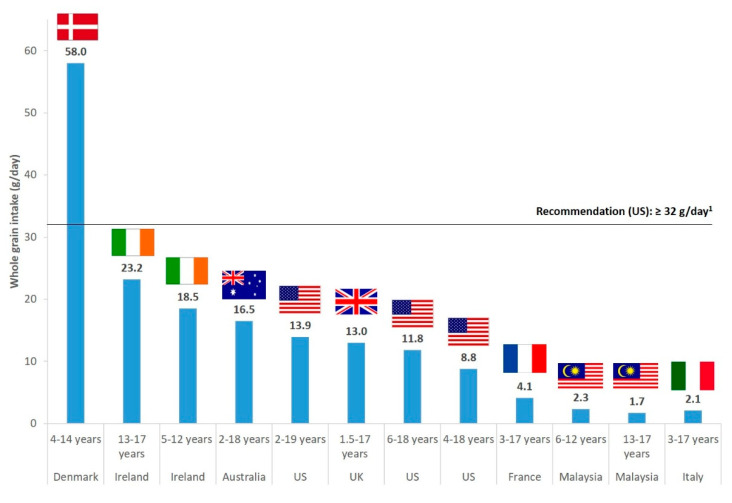
Whole grain daily intakes in children and adolescents in several countries as reported by nationally representative dietary surveys. Reported intakes are expressed in grams of whole grain ingredients and correspond to mean intakes in all cases, except for Australia and the United Kingdom (UK) where medians are reported. ^1^ Recommendation for whole grain consumption for children aged 3–7 years, as reported in the Dietary Guidelines for Americans 2015 (see Table 1 for details; recommendations for children energy intakes from the US Institute of Medicine have been used to convert recommendations from g/2000 kcal into g/day). Sources of the reported data are as follows: Denmark: Danish national survey of diet and physical activity 2011–2013 [18]; Ireland: 13–17 years: National Teens’ Food Survey (NTFS) 2005–2006; 5–12 years: National Children’s Food Survey (NCFS) 2003–2004 [19]; Australia: Australian Health Survey 2011–13 [20]; UK: National Diet and Nutrition Survey (NDNS) 2008–2011 [21]; US (United States): 2–19 years: National Health and Nutrition Examination Survey (NHANES) 2015–2016 [22]; 6–18 years: NHANES 2011–2012 [23]; 4–18 years: NHANES 2009–2010 [24]; France: Comportements et Consommations Alimentaires en France (CCAF) 2010 [25]; Malaysia: MyBreakfast study 2013 [26]; Italy: INRAN-SCAI 2005–2006 [27].

**Figure 2 nutrients-12-02217-f002:**
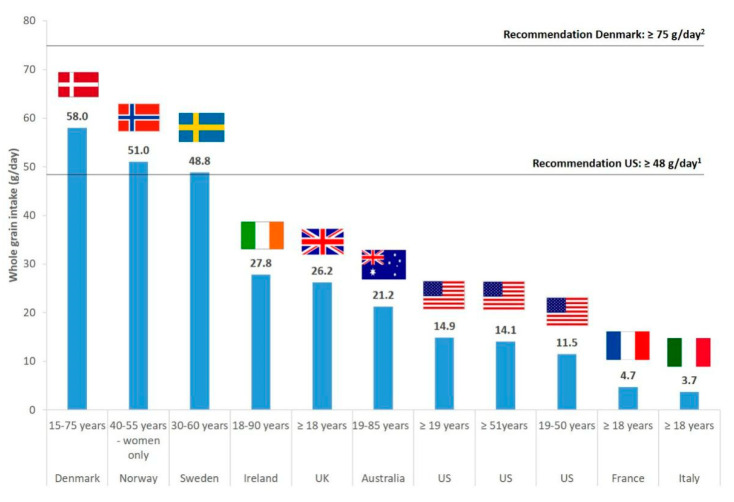
Whole grain daily intakes in adults in several countries as reported by nationally representative dietary surveys (except if specified otherwise). Reported intakes are expressed in grams of whole grain ingredients and correspond to mean intakes in all cases, except for Australia where medians are reported. ^1^ Recommendation for whole grain consumption for adults, as reported in the Dietary Guidelines for Americans 2015 (see Table 1 for details). ^2^ Recommendation for whole grain consumption for adults, as reported in the Danish dietary guidelines (see Table 1 for details). Sources of the reported data are as follows: Denmark: Danish national survey of diet and physical activity 2011–2013 [18]; Norway and Sweden: HELGA cohort subpopulations (not nationally representative; HELGA is a research project on Nordic health and whole grain food) [28]; Ireland: National Adult Nutrition Survey 2008–2010 [29]; UK: National Diet and Nutrition Survey (NDNS) 2008–2011 [30]; Australia: Australian Health Survey 2011–13 [20]; US (United States): ≥ 19 years: National Health and Nutrition Examination Survey (NHANES) 2015–2016 [22]; 19–50 and ≥ 51 years: NHANES 2009–2010 [24]; France: Comportements et Consommations Alimentaires en France (CCAF) 2010 [25]; Italy: INRAN-SCAI 2005–2006 [27].

**Figure 3 nutrients-12-02217-f003:**
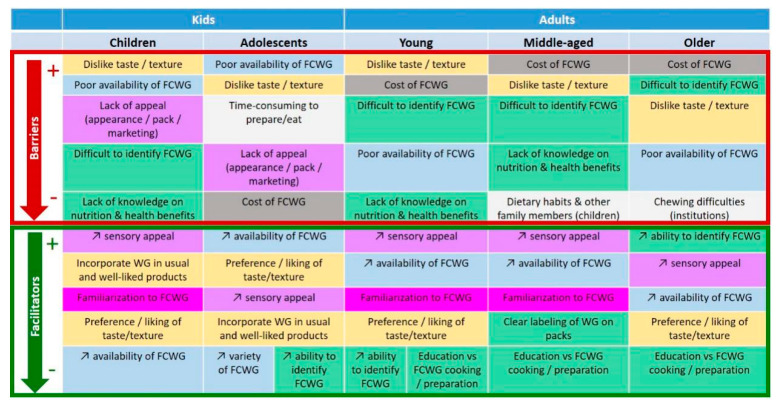
Main barriers to and facilitators of whole grain consumption in children and adults, as identified from the data collected in the frame of the current review. Factors are presented in decreasing order of their possible importance (from + to −), for each age group separately, on the basis of the number of studies that have concluded that the corresponding factor was a barrier or a facilitator. WG, whole grain. FCWG, foods containing WG.

**Figure 4 nutrients-12-02217-f004:**
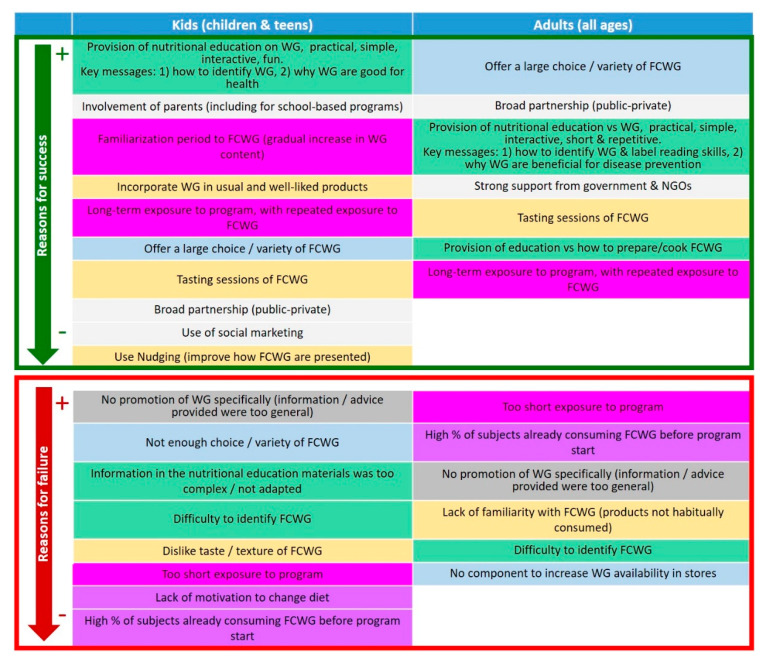
Main reasons for success or failure of programs that aimed at promoting whole grain consumption in children and adults, as identified from the data that were collected in the frame of the current review. Factors are presented in decreasing order of their possible importance (from + to −), for each age group separately, on the basis of the number of studies that have concluded that the corresponding factor was a reason for success or failure. NGOs: non-governmental organizations; WG: whole grain. FCWG, foods containing WG.

**Table 1 nutrients-12-02217-t001:** Recommendations regarding the consumption of whole grains as included in national dietary guidelines of several countries worldwide.

Country	Issuing Organization and Year	Age Range	Quantitative Recommendation (Recommended Quantities in Amounts of WG Ingredients)	Qualitative Recommendation (Statement)	Source of Identified Data
USA	USDHHS/USDA 2015 (DGA 2015)	Whole population ≥ 2 y	≥3 oz-eq ^1^/2000 kcal	Consume at least half of all grains as WG.	[5]
UK	PHE 2018	Whole population	None	Choose WG versions/varieties.	[6]
Brazil	Ministry of Health 2014	Whole population ≥ 2 y	None	Make natural or minimally processed foods the basis of your diet.	[7]
France	Santé publique France 2019	Adults	At least one WG starch per day (no information vs. corresponding quantity of WG ingredients)	Starches can be consumed every day. It is recommended to consume the WG version when they are grain-based: WG bread, WG rice, WG pastas, etc.	[8]
India	Indian National Institute of Nutrition 2011	Whole population	None	Use a combination of WG, grams (pulses) and greens. Increase consumption of WG.	[9]
Canada	Health Canada 2019	Whole population ≥ 2 y	None	WG should be consumed regularly. Eat plenty of WG food. Choose WG foods.	[10,11]
Denmark	Danish Veterinary and Food Administration 2013	Whole population	≥75 g/d	Choose WG first—it’s easy if you look for the WG logo when you shop.	[12]
Norway	Norwegian Directorate of Health 2014	Whole population ≥ 1 y	70–90 g/d	Eat WG cereal products every day.	[13]
Sweden	Swedish Food Agency 2015	Whole population ≥ 2 y	70 g/d in females—90 g/d in males	Choose WG varieties when you eat pasta, bread, grain and rice.	[14]

^1^ 1 oz-eq = 1 serving ≈ 16 g of WG ingredients (approximate and most often used equivalence). DGA, Dietary guidelines for Americans; oz-eq, ounce equivalent; PHE, Public Health England; USDHHS, US Department of Health and Human Services; USDA, US Department of Agriculture; WG, whole grain; y, years.

**Table 2 nutrients-12-02217-t002:** Description of programs that aimed at promoting the consumption of whole grains in children and adolescents, as identified in the frame of the current review.

Country	Program Name ^1^	Setting ^2^	Age Group(s) ^3^	Was the Program Focused on WG? ^4^	Components Used to Promote WG ^5^	Target(s) ^6^	Efficacy of the Program to Improve WG Intakes ^7^	Program Type ^8^	Study References
USA	WIC Food package (2009 revision)	Home	Children (preschoolers)	No	Federal aid program including provision of supplemental foods within the WIC Food package; since 2009, at least 50% of cereal products in the package are WG	Children & their mothers—low income	Neutral	Gov. (federal aid program)	[76,77,78]
USA	Power Plate	School	Older children	No	Use of emoticons + small prizes as incentives to select healthy foods such as a WG entrée	Children	Neutral	Academic	[75]
USA	Power of 3: Get healthy with WG foods	School	Older children	Yes	Classroom education lessons; school cafeteria menu changes to ↑ avail. of WG (for a variety of foods); family-oriented activities	Children & parents	Favorable	Academic	[74]
USA	FIT	School & community	Older children	No	Multicomponent initiative with school (e.g., interactive nutrition education, food tasting) & social marketing elements + community input—but nothing specific to WG	Children, parents, teachers, rep from community-based org. & school district	Favorable for WG bread; neutral for WG cereals (not further defined)	Mix (academic, public & private)	[79]
USA	CHANGE	School & community	Older children	No	Offering WG daily in school cafeteria menus + school education + parent & community components	Children, parents, teachers & school staff	Neutral	Private (NGO)	[81]
USA	Be a Fit Kid	School	Older children	No	Food tasting & nutrition education + parents meeting	Children & parents	Favorable	Academic	[82,83]
USA	Brauchla 2013	School	Older children	No (fiber)	Offering large variety of high-fiber snacks (mostly WG) at school	Children	Favorable	Academic & industrial	[80]
USA	Rosen 2008	School	Older children	Yes	Gradual incorporation of WG flour in bread products, burritos & chocolate chip cookies served at school meals	Children	Favorable	Academic & industrial	[71]
NL	Van Kleef 2014	School	Older children	Yes	Nudging (use of fun shape for WG breads)	Children	Favorable	Academic	[84]
USA	FUTP60	School	Adolescents	No	Social marketing + web-based support system for implementation of wellness policy—but nothing specific to WG	Children, teachers, school staff & parents	Favorable	Private	[87]
USA	Smarter lunch room	School	Adolescents with intellectual disabilities	No	Change from white to WG bread in peanut butter & jelly sandwiches + nudging (changes in how foods are presented & served)	Children	Favorable	Academic	[89]
USA	The Chef initiative	School	Adolescents	No	Training of school cafeteria staff with a professional chef to learn how to prepare healthy (incl. substitution of RG with WG) & palatable school lunches	Children & school food service staff	Inconclusive	Private (NGO)	[86]
USA	Radford 2014	School & home	Adolescents	Yes	Provision for FREE of a large variety of commercially available WG-containing foods at home and WG-containing snacks at school	Children	Favorable	Academic & industrial	[46]
Belgium	Aerenhout 2011	Home	Adolescents (athletes)	No	Non-stringent advice sent by email to consume more WG bread	Children & parents	Favorable for girls, neutral for boys (WG bread)	Academic	[85]
Finland	Hoppu 2010	School	Adolescents	No	Nutritional education as part of normal teaching and change in quality of snacks served at school—but nothing specific to WG	Children, parents, teachers, school food service staff, school heads	Favorable for girls, neutral for boys (rye bread)	Academic	[88]
UK	Rees 2010	School	Adolescents	No	Information leaflet tailored to each subject (according to answers to a baseline diet & psychological questionnaire)—no info vs. content related to WG	Children	Favorable for brown bread, neutral for WG breakfast cereals	Public (FSA)	[90]
USA	SNAP	Home	All ages	No	Federal aid program that provides money for food supply—but nothing specific to WG	Children & their families—low income	Neutral	Gov. (federal aid program)	[95,96]
USA	NSMP	School	Older children & adolescents	No	Before 2012: nothing; 2012–2014: 50% of grain foods served at school meals must be WG-rich (>50% WG ingredients); since 2014: 100%	School meal officers & children	Favorable following 2012 revisions; neutral before 2012 revisions	Gov. (federal aid program)	[91,92,93]
USA	Keast 2011	Food industries	Older children & adolescents	Yes	Modest change in food formulation to ↑ WG content in foods that are commonly consumed	Industrials	Favorable (modeling study only)	Industrial	[72]
Denmark	Fuldkorn (The Danish WG Partnership)	National campaign (home & food industries/bakers)	All ages	Yes	↑ in WG content of several commercial food products, use of WG logo on foods with high content in WG, communication to improve consumer knowledge vs. WG, information materials to assist bakers and retailers => broad partnership with involvement of multiple stakeholders	Overall population & industrials	Favorable (large ↗)	Mixed (gov. & industrial)	[18,97]
Greece	DIATROFI	School	All ages	No	Provision of WG-containing foods in school meals, nutritional education material distributed to children & families, health promotion events for children & parents (incl. chef demonstrations)	Children & parents from disadvantaged areas	Favorable	Academic	[94]

^1^ Full name or acronym of the program; if the program did not have a name, the reference of the corresponding study has been indicated here (name of first author and year of publication). ^2^ Environment where the program was implemented (e.g., at home, at school, etc.). ^3^ Children (aged 3–12 years) or adolescents (12–18 years). ^4^ A “no” answer would mean, for instance, that the program also aimed at promoting other types of foods or healthy behaviors in general. ^5^ Components of the program that were used to promote WG consumption. ^6^ The population that was the target of the program: it could be the subjects themselves or their care-givers. ^7^ Conclusion regarding the impact of the program on the consumption of WG by the study subjects, as measured throughout the study, with three possible conclusions: favorable (when the program was shown to induce a significant increase in the consumption of WG); neutral (no significant impact); inconclusive (when it was not possible to derive any clear conclusion from the study because of methodological issues, lack of information, or conflicting findings) (no study showed a detrimental impact, i.e., a significant decrease in the consumption of WG). ^8^ In relation to the entity (ies) instigating the program or that provided financial support for the research: governmental, industrial, academic (researchers), etc. Avail., availability; CHANGE, Creating Healthy, Active and Nurturing Growing-Up Environments; FSA, Food Standards Agency; FUTP60, Fuel Up to Play 60; Gov., governmental; NGO, non-governmental organizations; NSMP, National School Meal Programs (US National School Lunch and School Breakfast Programs); RG, refined grains; SNAP, Supplemental Nutrition Assistance Program (formerly called the “Food Stamp Program”); WG, whole grains; WIC, Food and Nutrition Service Special Supplemental Nutrition Program for Women, Infants, and Children.

**Table 3 nutrients-12-02217-t003:** Description of programs that aimed at promoting the consumption of whole grains in adults, as identified in the frame of the current review.

Country	Program name ^1^	Setting ^2^	Age & Sex (% Women) Group(s) ^3^	Was the Program Focused on WG? ^4^	Components Used to Promote WG ^5^	Target(s) ^6^	Efficacy of the Program to Improve WG Intakes ^7^	Program Type ^8^	Study References
Canada	Williams 2013	University	Young (college students) (both; 87%)	No	Introductory nutrition course (1 semester), with no particular focus on WG	College students	Inconclusive	Academic	[101]
USA	Ha 2011	University	Young (college students) (both; 88%)	No	Interactive introductory nutrition course (1 semester), focusing on disease prevention with 4h on WG-related topics (examples of interactive activities: food label hunt, WG tasting)	College students	Favorable	Academic	[49]
USA	Arts 2016	University	Young (college students) (both; 78%)	Yes (+ low-fat dairy)	Delivery of WG nutrition education messages on the Point-of Selection (POS) sites in dining halls and by email/text messages (daily during 3 wks) (and same for low-fat dairy)	College students	Neutral	Academic	[98]
USA	WIC Food package (2009 revision)	Home	Young (pregnant/post-partum women, mothers of 5y^-^children) (100%)	No	Federal aid program including provision of supplemental foods within the WIC Food package; since 2009, at least 50% of cereal products in the package are WG	Pregnant or post-partum women, or mothers of 5y- children	Neutral for objective measure, favorable for self-perceived consumption (6 & 18 mo after implementation of revision)	Gov. (federal aid program)	[76,99,100]
USA	Croy 2005	Not specified	Middle-aged (health club members, frequent consumers of WG) (100%)	Yes	Interactive educational program with four 60/90-min weekly meetings (topics/activities: health benefits of WG, info & strategies to identify WG & read labels, supermarket tour, WG bread tasting)	Adults	Favorable	Academic	[57]
USA	Power of 3: Get healthy with WG foods	Home & community	Middle-aged (parents of 9–11y children) (both; 91%)	Yes	Interactive educational program with family-oriented activities (weekly newsletters, classroom lessons vs. WG definition & identification, baking & grocery store tours with a “hunt for WG”, tasting of WG-containing foods) (over 5 mo)	Parents (and their children)	Neutral	Academic	[74]
USA	Is it Whole Grain?	Community ^9^	Older (community-dwelling) (both; 89%)	Yes	Interactive educational program with hands-on activities and tasting of WG-containing foods (three 1-h weekly sessions, with delivery of worksheets, informational handouts, recipes) (topics: WG definition & identification, WG health benefits)	Older adults themselves	Favorable	Academic	[63]
USA	Whole Grains and Your Health Program (part of the “Georgia Older Americans Nutrition Program”)	Institution (centers for seniors)	Older (resident in centers for seniors, and attending a congregate meal program) (both; 88%)	Yes	Educational program (5 lessons with delivery of handouts) (topics: WG definition & identification, WG health benefits, tips vs. how to include WG in the diet/cook WG) (1 to 2 lessons/mo over a period of 5 to 6 mo)	Older adults themselves	Neutral	Academic	[62]
Denmark	Fuldkorn	National campaign (home & food industries/bakers)	All ages (general population) (both; 53%)	Yes	↑ in WG content of several commercial food products, use of WG logo on foods with high content in WG, communication to improve consumer knowledge vs. WG, information materials to assist bakers and retailers => broad partnership with involvement of multiple stakeholders + organization of special campaigns targeted at specific groups (e.g., young men -> WG communication in cafes, night clubs…, and on internet and social media)	Overall population (including adults of all ages) & industrials	Favorable (large ↗ for all adults, and for men and women separately)	Mixed (gov. & industrial)	[18,97,102]
UK	WHOLEheart study^10^	Home	Young & middle-aged (healthy overweight adults with low habitual WG intakes)(both; 52%)	Yes	16-week WG familiarization period requiring participants to consume 60–120 g WG ingredients/d (a wide variety of WG-containing foods was provided free of charge to participants, from which they could self-select their preferred foods)	Adults	Favorable (elective WG consumption 1y after the end of the familiarization period)	Academic	[73]

^1^ Full name or acronym of the program; if the program did not have a name, the reference of the corresponding study has been indicated here (name of first author and year of publication). ^2^ Environment where the program was implemented (e.g., at home, at university, etc.). ^3^ Young adults (18–30/40 years), middle-aged adults (30/40–60/65 years) and older adults (> 60 years). ^4^ A “no” answer would mean, for instance, that the program also aimed at promoting other types of foods or healthy behaviors in general. ^5^ Components of the program that were used to promote WG consumption. ^6^ The population that was the target of the program; it could be the subjects themselves or their care-givers. ^7^ Conclusion regarding the impact of the program on the consumption of WG by the study subjects, as measured throughout the study, with three possible conclusions: favorable (when the program was shown to induce a significant increase in the consumption of WG); neutral (no significant impact); inconclusive (when it was not possible to derive any clear conclusion from the study because of methodological issues, lack of information, or conflicting findings) (no study showed a detrimental impact, i.e., a significant decrease in the consumption of WG). ^8^ In relation to the entity (ies) instigating the program or that provided financial support for the research: governmental, industrial, academic (researchers), etc. ^9^ Program sites included, but were not limited to, apartments for seniors, retirement communities, and centers for seniors. ^10^ The WHOLEheart study was a 16-week randomized controlled trial originally aimed at evaluating the impact of increased WG consumption on cardiovascular disease risk factors. Gov., governmental; mo, months; WG, whole grains; WIC, Food and Nutrition Service Special Supplemental Nutrition Program for Women, Infants, and Children; wks, weeks; y, year.

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
