# Peer review of "Main Factors Influencing Whole Grain Consumption in Children and Adults—A Narrative Review"

_nutrients, 2020, doi:10.3390/nu12082217_

Round 1
Reviewer 1 Report
Many papers have been published regarding the importance of and the factors influencing the whole grain consumption. Therefore, any paper that deals with whole grain consumption should show something innovative in order to contribute to an already large and interesting pool of literature.
The motivation for the study is not clear to the reader. Consequently the reader is left wondering what the purpose of the paper is. In my opinion, the reason for undertaking this research needs to be made more clear, while the contribution of the paper to the existing literature also needs to be added. These points are crucially important in order to justify such research being carried out.
The section of literature review lacks substantive discussion. As there is no substantive theoretical background, the presentation and the description of the whole grain consumption lacks a frame of reference and, as a result, cannot be put in context.
Most significantly, the discussion section does not incorporate and develop the findings of the literature review and the theory so as to provide the reader with a link between theory and practice. I feel that the paper does not contain any substantial discussion that develops and expands the theory in a substantive manner.
I found no mention of the limitations of the study, nor of the usefulness of the study to policy makers. The implications of the study are not stated so it is not clear to the reader how the conclusions of this research will facilitate the decision-making of policy makers towards more effective policies.
Author Response
Dear Reviewer,
We thank you very much for taking the time to read thoroughly our manuscript and for your very relevant and meaningful comments. Please find below (in red colour) the answers to your different comments. You will also find attached to the current resubmission an updated version of our manuscript in which we have strived to adequately address each comment satisfactorily (file named “nutrients-866240_Revised Manuscript”). As requested, changes in the revised manuscript are marked with the “Track Changes” option in Microsoft Word.
Please, note that all line numbers mentioned below refer to the file ‘nutrients-866240_Revised Manuscript’.
Point 1: Many papers have been published regarding the importance of and the factors influencing the whole grain consumption. Therefore, any paper that deals with whole grain consumption should show something innovative in order to contribute to an already large and interesting pool of literature.
The motivation for the study is not clear to the reader. Consequently the reader is left wondering what the purpose of the paper is. In my opinion, the reason for undertaking this research needs to be made more clear, while the contribution of the paper to the existing literature also needs to be added. These points are crucially important in order to justify such research being carried out.
Response 1: We acknowledge the reviewer’s comments and we have modified the introduction in order to describe more clearly the motivation of our study and how it would contribute to expand the existing literature. The amended sentences can be found in lines 46 to 60. Furthermore, the objectives of our review have also been described more clearly in lines 91 to 95 of the introduction. Finally, the discussion has also been enriched to provide more details regarding how the findings from our review articulate within the existing literature and what these findings bring to the topic (see added sentences from lines 356 to lines 390).
Point 2: The section of literature review lacks substantive discussion. As there is no substantive theoretical background, the presentation and the description of the whole grain consumption lacks a frame of reference and, as a result, cannot be put in context.
Most significantly, the discussion section does not incorporate and develop the findings of the literature review and the theory so as to provide the reader with a link between theory and practice. I feel that the paper does not contain any substantial discussion that develops and expands the theory in a substantive manner.
Response 2: The discussion has been enriched in order to further discuss and put in context the findings of our review and to better relate theory and practice. Corresponding added sentences can be found in lines 356 to 368 and in lines 372-390 (the latter sentences have also been used to address reviewer’s point 4 below).
Point 3: I found no mention of the limitations of the study.
Response 3: We have added a whole paragraph in the discussion to address the limitations (and strengths) of our study (see lines 396 to 410 for limitations, and lines 391 to 396 for strengths).
Point 4: I found no mention of the usefulness of the study to policy makers. The implications of the study are not stated so it is not clear to the reader how the conclusions of this research will facilitate the decision-making of policy makers towards more effective policies.
Response 4: We have added a whole paragraph in the discussion to better describe the usefulness of our review to policy makers and the possible implications of our review in regard to policy making (see lines 372 to 390).
Reviewer 2 Report
This study reviews the literature on factors that influence whole grain consumption in adults and children. Five methods to facilitate wholegrain intake are identified:
- Increase the availability and variety of foods with WG.
- Improve the appeal of WG food.
- Reduce the cost of WG food.
- Gradually increase consumer exposure to WG foods.
- Make it easy for consumers to identify WG foods.
These are all sound recommendations, but not entirely surprising.
I do not think that this list alone fills any gaps in our knowledge. The reason is, apply the list of methods above to any consumer product and one will expect an increase in consumption. These are things that make any good more appealing, not just WG foods.
The value added of this review comes from what can be said about the relative importance of the different methods in different contexts, in this case age groups of the consumers (Figure 3). This needs to be emphasised.
The reader needs to be provided with greater clarity with respect to how the different barriers and facilitators in Figure 3 differ from one another. For example, how does “sensory appeal” differ from “preference/liking of taste/texture” ?
I am sceptical about how useful is some of the information in Figure 4. For example, how does “Broad partnership” lead to the success of a WG program? Broad partnership must work only if it facilitates increasing one of the 5 items above. This is not clearly explained.
The methods used to draw conclusions in the paper seem a little adhoc. It would beneficial to the reader to include a clear description of the scientific approach.
Does the literature have anything to say about how much the above methods contribute to the extreme differences in WG consumption we observe in different counties (figures 1 and 2)? Are WGs inexpensive in Denmark and expensive in Malaysia? Surely the reasons for low consumption of WGs are dependent on context.
Author Response
Dear Reviewer,
We thank you very much for taking the time to read thoroughly our manuscript and for your very relevant and meaningful comments. Please find below (in red colour) the answers to your different comments. You will also find attached to the current resubmission an updated version of our manuscript in which we have strived to adequately address each comment satisfactorily (file named “nutrients-866240_Revised Manuscript”). As requested, changes in the revised manuscript are marked with the “Track Changes” option in Microsoft Word.
Please, note that all line numbers mentioned below refer to the file ‘nutrients-866240_Revised Manuscript’.
Point 1: This study reviews the literature on factors that influence whole grain consumption in adults and children. Five methods to facilitate wholegrain intake are identified:
- Increase the availability and variety of foods with WG.
- Improve the appeal of WG food.
- Reduce the cost of WG food.
- Gradually increase consumer exposure to WG foods.
- Make it easy for consumers to identify WG foods.
These are all sound recommendations, but not entirely surprising.
I do not think that this list alone fills any gaps in our knowledge. The reason is, apply the list of methods above to any consumer product and one will expect an increase in consumption. These are things that make any good more appealing, not just WG foods.
Response 1: We thank the reviewer for this important comment. A sentence has been added in the discussion to address this issue (lines 338 to 340).
Point 2: The value added of this review comes from what can be said about the relative importance of the different methods in different contexts, in this case age groups of the consumers (Figure 3). This needs to be emphasised.
Response 2: Substantial changes have been applied to section 2 of the review to address the reviewer’s comment. More precisely, three sentences have been added in lines 156-162 and several existing sentences have been modified (lines 183-192) in order to emphasise the relative importance of the different identified methods depending on consumers’ age groups, and to highlight the similarities and the key differences observed between the different age groups.
Point 3: The reader needs to be provided with greater clarity with respect to how the different barriers and facilitators in Figure 3 differ from one another. For example, how does “sensory appeal” differ from “preference/liking of taste/texture” ?.
Response 3: Several sentences have been added in section 2 (lines 121-138) to address the reviewer’s comment. These sentences aimed to better define the different barriers and facilitators that have been identified, and how they differ from one another. This has been achieved for the two examples provided by the reviewer as well as for several other factors (actually all factors that we considered as being possibly confusing).
Point 4: I am sceptical about how useful is some of the information in Figure 4. For example, how does “Broad partnership” lead to the success of a WG program? Broad partnership must work only if it facilitates increasing one of the 5 items above. This is not clearly explained.
Response 4: A whole paragraph has been added in section 3 (from line 290 to line 311) in order to explain more clearly how “broad partnership” could lead to the success of a WG program.
Point 5: The methods used to draw conclusions in the paper seem a little adhoc. It would beneficial to the reader to include a clear description of the scientific approach.
Response 5: In order to address the reviewer’s comment and to better describe the scientific approach used for this research, a total of four sentences have been added in section 2 (two sentences within lines 116 to 121 and two sentences within lines 138 to 142), and several sentences have been modified and/or added in section 3 (within lines 196 to 198 and lines 206 to 217).
Point 6: Does the literature have anything to say about how much the above methods contribute to the extreme differences in WG consumption we observe in different counties (figures 1 and 2)? Are WGs inexpensive in Denmark and expensive in Malaysia? Surely the reasons for low consumption of WGs are dependent on context.
Response 6: Several sentences (from lines 356 to 368) have been added in the discussion to mention the information available in the literature regarding how much the “above methods” contribute to the extreme differences in WG consumption observed in different counties, focusing in particular on the example of Denmark (which is the country with the highest level of WG consumption among all countries that have been considered in our review).
Regarding the second aspect which has been raised by the reviewer (“Are WGs inexpensive in Denmark and expensive in Malaysia”), the question of the comparative cost of WG-containing products in different countries around the world would be very interesting to address, and we deeply thank the reviewer for his/her valuable comment. However and unfortunately, we do not know if this type of information is currently available, and it was not possible for us to gather this information in a sufficiently accurate manner in the frame of the review process of our manuscript.